# Effect of Previous Crops and Soil Physicochemical Properties on the Population of *Verticillium dahliae* in the Iberian Peninsula

**DOI:** 10.3390/jof8100988

**Published:** 2022-09-21

**Authors:** Antonio Santos-Rufo, Mario Pérez-Rodriguez, Juan Heis Serrano, Luis Fernando Roca Castillo, Francisco Javier López-Escudero

**Affiliations:** 1Excellence Unit ‘María de Maeztu’ 2020-23, Department of Agronomy, Campus de Rabanales, University of Cordoba, 14071 Cordoba, Spain; 2Department of Agroforestry Sciences, ETSI University of Huelva, 21007 Huelva, Spain

**Keywords:** cation exchange capacity, Guadalquivir Valley, history crops, neural network, *Olea europaea*, soil texture, tree decision, Verticillium wilt of olive

## Abstract

The soil infestation of *Verticillium dahliae* has significant Verticillium wilt of olive (VWO) with epidemiological consequences which could limit the expansion of the crop. In this context, there is a misunderstood history of the crops and soil property interactions associated with inoculum density (ID) increases in the soil. In this study, the effect of the combination of both factors was assessed on the ID of *V. dahliae* in the olive-growing areas of the Iberian Peninsula. Afterwards, the relationship of the ID to the mentioned factors was explored. The detection percentage and ID were higher in Spain than Portugal, even though the fields with a very favourable VWO history had a higher ID than that of the fields with a barely favourable history, regardless of the origin. The soil physicochemical parameters were able to detect the degree to which the ID was increased by the previous cropping history. By using a decision tree classifier, the percentage of clay was the best indicator for the *V. dahliae* ID regardless of the history of the crops. However, active limestone and the cation exchange capacity were only suitable ID indicators when <2 or 4 host crops of the pathogen were established in the field for five years, respectively. The *V. dahliae* ID was accurately predicted in this study for the orchard choices in the establishment of the olive.

## 1. Introduction

*Verticillium dahliae* Kleb. is a widespread soil-borne pathogen causing Verticillium wilts in many economically significant crops in temperate zones [1]. In the world’s main olive growing areas, located in the Mediterranean Basin, such as those in the south of the Iberian Peninsula, this pathogen causes Verticillium wilt of olive (VWO), the most destructive fungal disease affecting this strategic crop for this geographical region [2,3]. In particular, in the Guadalquivir Valley (Jaen, Cordoba, and Seville Spanish provinces), the major cause of disease spread and its increase in recent decades is closely associated with the cultivation of the hosts of the pathogen (mainly cotton and vegetables, such as tomato, potato, eggplants, etc.) during the years previous to the establishment of olive orchards [4,5].

In fact, the infestation of the soil with microsclerotia (the infective and survival structures of the pathogen in the soil; MS) is the most restrictive factor for spreading new olive orchards, and its level has been strongly correlated with the disease incidence and severity of the epidemics [6]. In this context, Spanish plant health services generally warn farmers that before planting olive trees, it is highly recommended to carry out a soil analysis of the population of *V. dahliae* in the soil and to investigate what the history of crops has been for at least the previous 10 years, due to the prolonged survival of the MS in the soil. However, the information available in this regard is limited, although there are some investigations aimed at determining the effect of cropping history on the densities of *V. dahliae* and in the management of Verticillium wilt in hosts such as cotton [7,8] or potato [9].

On the other hand, the progress of the incidence of VWO in the years after planting seemed to also depend on the type of soil [4]. Nevertheless, there are few studies that have addressed the influence of edaphic factors, such as pH, cation exchange capacity (CEC), or soil texture, on *V. dahliae* in the soil or on the disease. It is known that *V. dahliae* normally survives in alkaline or neutral soils (pH between 6 and 9), but when the pH drops below 5.5, the survival of the pathogen and the production of MS decreases [1]. Another soil characteristic associated with the suppression or development of *V. dahliae* is the presence of high levels of active limestone that directly affects the CEC and the pH. On the other hand, Ca and Mg, two of the most influential elements in the CEC, are believed to play a key role in disease resistance as they aid in cell wall formation, increasing its resistance to attack by numerous enzymes and preventing pathogen penetration, as reported by Pennypacker (1989) [10]. Regarding this matter, recent research showed that, in the case of olive plants, treatments with N and Na excess, and with N and Na excess and K deficiency, decreased the germination of the fungus MS and the disease progress after artificial inoculations [11]. Soil texture also appears to have a variable effect on the pathogen, probably due to the complexity of existing interactions [1]. In olive, field surveys that covered the main olive producing areas in the south of the Iberian Peninsula showed that the disease incidence was significantly lower (mean percentage 12.9%) in olive orchards established in Alfisol than that registered in plantations established in Entisol, Inceptisol, or Vertisol soils [4]. Soil density also seems to influence VWO since those with the highest values tend to be cooler soils that have optimum temperatures for the development of the disease in summer. In sandy soils, the frequency of irrigation is higher than in heavy soils; so, the effect of irrigation on the disease occurrence is increased [1].

The effect of the interaction of both factors, the history of the crops and the soil properties, on the inoculum density (ID) of *V. dahliae* and VWO progress is likely to be complex and could lead to misinterpretation of the results as focusing on only one parameter causes the others, which also simultaneously influence the ID, to be ignored [1,12]. In this sense, diverse machine learning algorithms have been successfully applied for studying some plant diseases. For instance, Ghini et al. (2016) [13] applied a logistic model for studying the stalk rot of corn, in which multiple biotic (*Fusarium* spp.) and abiotic (pH, organic matter, P, K, Ca, Mg, or CEC) soil factors were involved. Moreover, tree decision algorithm analyses have been satisfactorily used for VWO regarding soil temperature [14] or the water irrigation effect [15]. Therefore, similar methodologies could be appropriate for the study of the relationship between the history of the crops, the soil properties, and the ID of *V. dahliae* in the olive-growing areas of the Iberian Peninsula.

Therefore, the objectives of this study were to model (i) the effects of the crops previous to the establishment of the olive orchards on the ID of *V. dahliae* in the olive-growing areas of the Iberian Peninsula, with an assessment of the physicochemical characteristics from the soil and (ii) how the ID of *V. dahliae* is related to the levels of these physicochemical parameters.

## 2. Materials and Methods

For this study, the data on the ID of *V. dahliae* and the physicochemical properties of a wide collection of soil samples collected during the last 20 years in areas of the Iberian Peninsula were used as explained below. In these soils, new olive plantations were to be established. Soil analyses of the ID were carried out by the Diagnostic Service and Phytopathological Analyses (DSPA) of the Department of Agronomy of the University of Córdoba (Units of Excellence Maria de Maeztu); concurrently, physicochemical analyses of the soil were performed by public or private laboratories, mainly by the Laboratorio Agroalimentario of the Join of Andalusia (Córdoba). 

The soil samples came from zones of the peninsula where new olive orchards were to be planted, mostly from Spain and southern Portugal. The geographical origin of the soil samples is detailed in Figure 1. The number of samples analyzed amounted to 59 in Andalusia, 43 of them corresponding to the Guadalquivir Valley; another group of 8 samples corresponded to the areas of Castilla la Mancha, Castilla León, Murcia, and Extremadura Region; and finally, a total of 17 samples were received from Portugal.

The agronomic characteristics of the surveyed fields with regard to the history of the cultivated species during the years previous to the soil analyses are shown in Table 1. Four groups were established in relation to the presence of host crops of *V. dahliae* (Figure 1 and Table 1) in the fields in the five years previous to the soil analyses: (1) VWO with barely favourable previous cropping history, which had <2 host crops of *V. dahliae* in the previous five years; (2) VWO with favourable previous cropping history, which had 3 host crops of the fungus; (3) VWO with very favourable previous cropping history, which indicated that only one of the previous five years of the plot was free of *V. dahliae* hosts; and (4) VWO with extremely favourable previous cropping history, which indicated that during all of the five previous years species susceptible to *V. dahliae* were cropped in the plot.

Soil sampling was carried out in most cases by the owners of the farms or by the technicians of the agronomic services companies, according to the methodology proposed by López-Escudero et al. (2003) [16], uploaded at the DSPA web page (https://www.uco.es/agronomia/es/transferencia/servicios-de-transferencia/servicio-de-diagnostico-fitopatologico, accessed on 27 March 2022) and summarized in Figure 2.

Each of the samples consisted of different subsamples that were taken every 10–20 m on the longest diagonal of each of the subplots. A cylindrical soil auger was used for sampling. Each subsample consisted of 100–200 g of soil taken at a depth of between 5 and 30 cm once the surface layer (5 cm of soil) was removed. These subsamples were mixed at the end of the process, giving rise to the samples (Figure 2).

### 2.1. Quantification of Inoculum Density of Verticillium dahliae in Soil Samples

Details of the soil assays for *V*. *dahliae* are provided in Roca et al. (2015) [5]. Briefly, the soil was air-dried at room temperature for 30 days and then sieved (0.8 mm diameter) to eliminate big particles. Then, the ID of *V. dahliae* in each soil sample was estimated by wet sieving [17] by splitting each soil sample across 10 plates of a modified sodium polypectate agar medium (MSPA) [18]. The MSPA plates were incubated for 14 days at 24 °C in the dark. After incubation, soil residues were removed by cleaning the plate surfaces under tap water, and the colonies of *V. dahliae* were counted under a stereoscopic microscope (Nikon SMZ-2 T, Tokyo, Japan). The ID in each soil sample was estimated from the number of *V. dahliae* colonies and expressed as MS per gram of air-dried soil (MS/g).

### 2.2. Physicochemical Properties of Field Soils

The physicochemical parameters determined in the soil analyses were: cation exchange capacity (CEC; meq 100g^−1^); Ca exchange (Ca; meq 100g^−1^); Mg exchange (Mg; meq 100g^−1^); Na exchange (Na; meq 100g^−1^); K exchange (K; meq 100g^−1^) (determined by the technique of inductively coupled plasma combined with optical emission spectroscopy—ICP-OES); assimilable P (P; meq 100g^−1^) (determined by UV/Vis spectrophotometry); active limestone (lime; %) (determined by potentiometry); organic matter (OM; %) (obtained by the K_2_Cr_2_O_7_-H_2_SO_4_ oxidation method); pH (using a pH meter to measure the pH of a 1:2.5 soil: water suspension); and texture (percentage of clay, sand, and silt according to USDA Soil Taxonomy, determined by the pipette and sieve method; clay, sand, and silt parameters).

### 2.3. Data Analysis for Assessing the Effect of Soil Properties and Crop Rotation on Explored Fields on the Inoculum Density of Verticillium dahliae

The overall effect of the experimental treatment combinations on the ID of *V. dahliae* and the physicochemical parameters was explored to determine the amount of variability explained by the study factor (i.e., the previous cropping history group), using mixed-effects models. The experiments and blocks were considered as random effects. The previous cropping history groups and their interactions were considered to be fixed effects. Tukey’s honestly significant difference test was used for all pair-wise comparisons among the treatments.

The overall response of the experimental treatment combinations to the ID of *V. dahliae* CEC, Ca, Mg, Na, K, P, lime, OM, pH, and clay, sand, and silt parameters was first explored by cluster analyses. The purpose of these analyses was to establish functional groups of correlated experimental treatments. Then, the relationship between the classes of ID and the mentioned physicochemical parameters was estimated using 25 different linear and nonlinear machine learning algorithms, with the aim of assessing their robustness: multilayer perceptron, stochastic gradient descent classifier, passive-aggressive classifier, linear support vector classifier, calibrated classifier CV, ridge classifier CV, ridge classifier, label propagation, label spreading, logistic regression, linear discriminant analysis, support vector machines, extra trees classifier, k-neighbors classifier, Gaussian naive Bayes, bagging classifier, random forest classifier, nearest centroid, light gradient boosting machine classifier, extra tree classifier, Bernoulli naive Bayes, extreme gradient boosting classifier, decision tree classifier, adaptive boosting classifier, quadratic discriminant analysis, and dummy classifier. These algorithms were fitted to the abovementioned parameters as explanatory variables, and the ID class was the response variable, taking the lowest ID class as the reference category. The dataset was partitioned 75%–25% into training and test sets, and 10-fold cross-validation was used to estimate the accuracy of the models, with precision as a parameter for scoring. The classifiers that learned the best were then evaluated on the test set and used to identify the groups that were homogeneous in terms of *V. dahliae* ID. The results were summarized as a final accuracy score, a confusion matrix, and a classification report, which are the common performance measurements for the machine learning classification of several plant diseases [19].

The mixed-effects models and cluster analyses were conducted in R (http://www.R-project.org, accessed on 23 March 2022) with the packages lme4 [20] and cluster [21], respectively. The machine learning algorithms were simultaneously implemented using version 0.19.1 of the Python (https://www.python.org/psf/, accessed on 23 March 2022) library Lazy predict (https://pypi.org/project/lazypredict/, accessed on 23 March 2022).

## 3. Results and Discussion

### 3.1. Inoculum Density of Verticillium dahliae in Soil

The results of the quantitative analyses of the population of *V. dahliae* in the soil were evaluated in samples taken from plots where olive plantations were to be established. These samples (84) were sent to the Diagnostic Service of the Agroforestry Pathology group of the University of Córdoba during the years 2005 to 2016 from Andalusia and other communities in Spain, as well as from Portugal. Thus, the origin of the samples is very variable, comprising very different types of soil with respect to their edaphic characteristics, the microbial flora, the crops, the plant species that have grown in them, etc.

The highest detection percentage and ID were found in Spain (67.8% for Andalusia, with a mean value of 7.0 ± 11.9 ppg; and 25.0% for the rest of Spain, with a mean value of 0.8 ± 2.1 ppg); these values were much lower in the analyzed Portuguese soils, where the pathogen was detected in only 17.6% of the samples, with a mean value of 0.1 ± 0.2 ppg. This fact is surely because the new plantations in Portugal are being established on non-infested areas that come from clearing *dehesas* [22], with poor soils which are relatively more acidic than those in the Andalusian areas and in which the pathogen is not usually present. In Andalusia, 35.6% (21 samples) of the soils analyzed in the Guadalquivir Valley have an ID in the range of 0.4 to 6.8 ppg. This is an amount that should be considered moderately high, particularly in the case of the highly virulent defoliant isolates in which the disease has been shown to occur, causing very high incidences of up to 55% in the Picual cultivar 3 years after planting, where it exceeds 3.0 ppg [6]. In this sense, it must be considered that in Spanish areas outside of Andalusia or in Portugal the defoliant isolates are probably not as present as in the Guadalquivir Valley, where the disease pressure (inoculum potential) is very high and has shown a clear prevalence of the most virulent isolates (defoliant pathotype) [23].

In summary, it can be indicated that the pathogen is distributed in a large part of Spain and Portugal, associated with soils in which olive groves have been established in recent years or in olive groves established in recent decades. The samples sent for analysis came in most cases from areas where the owners considered that they were in a place of risk, due to the previous cultivation of hosts, the influence of neighboring crops, or the presence of the disease in olive groves in the area. All these aspects have a considerable influence not only on the presence and quantity of the pathogen in the soil, but also on its detection since the main quantification methods are based on sample screening procedures and culture media which are used with different compounds and salts that can interact with the physical and chemical characteristics of the samples. This variability clearly influences the mean values presented for each area, although we believe that it can be replaced by the large volume of analyses carried out, which have made the sampling representative.

Regardless of the origin, the type III tests of the fixed effects from the mixed-model analysis showed significant (*p* < 0.05) effects of the previous cropping history group on the ID of *V. dahliae* in soils from the Iberian Peninsula (data not shown and Figure 3a), as has been reported in other studies [7,8,9,24,25]. The *V. dahliae* ID was, by mean, ten times higher when cropping susceptible hosts to *V. dahliae* for five, or for four out five, of the previous years (very and extremely favourable treatments; mean of 12.4 ± 14.4 and 11.0 ± 14.9 Ms/g, respectively) than when cropping < 2 or 3 susceptible species to *V. dahliae* during the previous five years (barely favourable and favourable treatments; mean of 0.2 ± 0.6 and 3.0 ± 4.4 Ms/g, respectively) (Figure 3a). In general, the fields repeatedly cropped with a susceptible crop such as cotton or potato had a higher ID, as seen in the ‘Galan (gasolinera)’ and ‘Juan Carlos Roldan Zona 3’ fields in this study (46.6 and 52.0 Ms/g, respectively; Appendix A) and other studies [7,8,9,24,25]. However, there are fields, such as *Antonio Valle* (*Suelo negro*) in Cordoba (Spain), with <2 host crops to *V. dahliae* and an ID of 2.4 Ms/g. Similar results were reported by Huisman and Ashworth (1976) [24] in some fields in the San Joaquin Valley of California (USA), where the *V. dahliae* ID increased rapidly following only 1 year of susceptible crops. In contrast, *V. dahliae* was not detected in some fields in Vejer de la Frontera (Andalusia, Spain) which had been repeatedly cropped with a susceptible crop (e.g., ‘Las Lomas P-42–4’ field, Appendix A), but this aspect needs to be elucidated by further research.

### 3.2. Physicochemical Parameters

The soil physicochemical parameters significantly varied among the assessed fields with different cropping histories in the Iberian Peninsula. The previous cropping history effects were significant (*p* < 0.05) for the Mg exchange (Mg; meq 100g^−1^), Na exchange (Na; meq 100g^−1^), and soil texture (clay, sand, and silt; %) parameters (data not shown and Figure 3d,e,k–ll).

The type III tests of the fixed effects from the mixed-model analysis showed that Mg varied significantly between 4.6 ± 3.2 and 2.4 ± 1.3 meq 100g^−1^, with the lowest and the highest values observed from the barely and very favourable previous cropping history treatments, respectively. In addition, the highest Na (1.4 ± 1.2 meq 100g^−1^) and clay (43.3 ± 18.0%) contents were observed from the very favourable treatment, whereas the lowest Na (0.4 ± 0.2 meq 100g^−1^) from the extremely favourable treatments and clay (29.0 ± 13.4%) from the barely favourable treatment were observed (Figure 3e,k). The sand parameter was significantly higher in the barely and extremely favourable treatments (mean of 50.7 ± 20.9 and 51.9 ± 19.4%, respectively) than in the favourable treatment (mean of 34.7 ± 18.1%), but the very favourable treatment reached a mean value (36.5 ± 19.4%) that did not differ between them (Figure 3l). Conversely, the silt parameter was significantly higher in the favourable (mean of 26.8 ± 10.6%) than the extremely favourable treatment (mean of 18.4 ± 9.5%), but the values from the barely favourable (mean of 20.4 ± 10.9%) and the very favourable (mean of 20.1 ± 8.5%) treatments did not differ between them (Figure 3ll).

The previous cropping history with the *V. dahliae* host and non-host affected the ID of the pathogen in the soil not only through its direct influence, but also through its influence on the physicochemical properties of the soil. The Mg exchange values were higher in the fields with the very favourable history than the barely favourable previous cropping history fields in the Iberian Peninsula, as well as in the fields with spring cotton (one of the most common host crops of *V. dahliae* in the favourable treatment) compared to spring maize single-cropping patterns for ten years in North China [26]. In addition, the Na exchange increased in the fields when the hosts of *V. dahliae* were cropped for four out of the five previous years (favourable treatment) in this study and when sugar beet (a host crop of *V. dahliae* employed in the very favourable cropping history group of this study) was intercropped with maize for five consecutive years in the Dura sub-catchment, northern Ethiopia [27].

Except for the extremely favourable treatment, with most of the fields corresponding to those from Vejer de la Frontera (Andalusia, Spain) where the soil is sandy loam, the percentage of clay tended to decrease with the decrease in the number of hosts of *V. dahliae* cropped in the previous five years (Figure 3a). This could be explained because cotton, one of the most common host crops of *V. dahliae* in the favourable treatment, is normally cropped in clayey soils and, in general, given its high profitability in Andalusia compared with other crops (where this treatment is mostly located), is not intercropped. However, there are studies indicating a percentage of clay values higher in *V. dahliae* non-host rotations, such as onion–maize–onion, than those with maize and sugar beet intercropping [27], which is a host of *V. dahliae* [1]. This could also explain that the percentage of sand was higher in the favourable than in the extremely favourable treatments in this study, as well as in a mono-cropping system of *Eragrostis tef* in the study carried out by Tesfahunegn and Gebru (2020) [27] in northern Ethiopia.

From the rest of the studied parameters (cation exchange capacity (CEC), Ca exchange (Ca), K exchange (K), assimilable P (P), active limestone (Lime), organic matter (OM), and pH), it was not possible to find differences (*p* > 0.05) between the barely favourable, favourable, very favourable, and extremely favourable treatments (data not shown and Figure 3b,c,f–j). This could be attributed to the fact that CEC, Ca, K, P, Lime, OM, and pH are the soil physicochemical parameters that are little influenced by the previous cropping history and its soil management practices. In general, it is reported that these parameters usually increase when cropping the non-hosts of *V. dahliae*, such as legume, onion, or grasses [28,29]. However, the highest and lowest CEC (from 20.1 to 30.2 meq 100g^−1^), Ca (from 16.3 to 23.2 meq 100g^−1^), and K (from 0.8 to 1.1 meq 100g^−1^) values were observed from the very and barely favourable treatments, respectively, as occurred for the significantly affected Mg, Na, and clay parameters. The values of P (32.4 to 60.7 meq 100g^−1^), lime (3.6 to 6.0 %), OM (1.4 to 1.5 %), and pH (7.9 to 8.3) did not follow any understandable trends. Such lack of influence could be attributed to the duration of the cropping system, the variability in the management practices and weather conditions, and the effects of variation in the topography.

### 3.3. Relationships among ID of V. dahliae in Soil and Physicochemical Parameters

The multivariate hierarchical cluster analysis yielded three—A to C—functional groups among the experimental treatments (four previous cropping history groups), which were associated with low (0–24.4, mean of 2.75 Ms/g), medium (0–52.0, mean of 5.4) Ms/g), and high (0–44.4, mean of 18.3 Ms/g) ID classes (Figure 4).

This analysis confirmed the results of the mixed-model analysis, highlighting a clear effect of the previous cropping history groups. In this sense, group A comprised six experimental treatments with a high ID level of *V. dahliae*, mostly including the very favourable treatment. In general, the treatments with a high ID level of *V. dahliae* were associated with high CEC, Ca, Mg, Na, K, and clay values, intermediate to high OM values, intermediate values of lime and pH, and low values of the P and sand and silt parameters. On the other hand, group B included 34 experimental treatments with a medium ID level of *V. dahliae* and high value for clay and silt. It included the barely favourable and favourable treatments and just three and seven very and extremely favourable treatments, respectively. This group of treatment combinations exhibited intermediate values for lime, OM, and pH, intermediate to low levels of CEC, Ca, Mg, Na, K, and P, and low levels for the percentage of sand. Finally, group C comprised 44 experimental treatments associated with a low ID level of *V. dahliae* and a high value of the percentage of sand. The treatments included mostly the barely and extremely favourable treatments and three and four favourable and very favourable treatments, respectively. The low ID level of *V. dahliae* and a high value of the percentage of sand observed in the treatment combinations within this group were associated with low levels of CEC, Ca, Mg, Na, K, lime and percentage of clay and silt. Intermediate values of P, OM, and pH were exhibited for this group of samples (Figure 4).

To the best of the authors’ knowledge, the soil physicochemical properties and cropping systems relationship has not previously been assessed in relation to the ID of *V. dahliae* in the soil. One of the most significant results of this study is that leaving only one of the previous five years of the plot free of *V. dahliae* hosts (VWO with a very favourable previous cropping history treatment) is related to high levels of the base cations Mg and Na (one of the most influential elements in the CEC) and percentage of clay and low levels of P and percentage of sand and silt, among others, which result in high levels of *V. dahliae* in the soil. The high ID levels in the soils with high Mg and Na contents that were mostly cropped with host crops of *V. dahliae* may be explained by a negative impact of these elements on the disease resistance of the hosts, favouring the disease and thus the dispersion of the pathogen in the soil. However, it is believed that Mg aids in cell wall formation, increasing its resistance to attack by numerous enzymes and preventing pathogen penetration [10]. This hypothesis is in contrast with the results found in this study, which are in line with the results found in other pathosystems, such as corn blight caused by the fungus *Bipolaris maydis* [30]. In this case, however, the high levels of Mg increase the severity of the disease. In addition, the high levels of *V. dahliae* in the soils with a high percentage of clay which were mostly cropped with host crops of *V. dahliae* may be because the MS could also be somehow more attached to the soil matrix by means of an electrostatic charge, as has been suggested for certain phytopathogenic viruses or bacteria [31]. This hypothesis could explain the ID differences found between previous cropping history treatments with different percentages of clay values in this study, as well as the higher disease incidence found in a disease survey conducted to assess the association of agronomical and geographical factors with the current spread of the VW in the Guadalquivir Valley, mostly established in the Entisol and Vertisol soils or presumably the soils with a high CEC and percentage of clay [4].

The other important results extracted from the multivariate analyses performed in this study are that the soil pH and the levels of P and OM are not key parameters to monitor the effects of the different cropping histories on the ID of *V. dahliae* in the soil. The lack of P and OM influence found in this study is unclear. To the best of the authors’ knowledge, the effects of P and OM on *V. dahliae* have not previously been quantified. However, the lack of a soil pH influence could be explained because the values from this study did not drop below 5.5, the pH value where the survival of the pathogen decreases, as has been cited [2].

Between the 25 different linear and nonlinear machine learning algorithms assessed with the training set (75%) from a dataset (84 samples) comprising all experimental combinations of this study, the multilayer perceptron yielded the highest accuracy, balanced accuracy, and F1 score (greater than 95.0, 88.0, and 94.0%, respectively) in classifying ‘low’, ‘medium’, and ‘high’ *V. dahliae* ID classes (data not shown and Table 2). The next group of algorithms comprised the decision tree classifier along with other algorithms, such as the support vector machines, with 4.9, 3.7, and 5.1% lower mean accuracy, balanced accuracy, and F1 score than the multilayer perceptron, respectively, followed by linear discriminant analysis, Bernoulli naive Bayes, logistic regression, the random forest classifier, the adaptive boosting classifier, quadratic discriminant analysis, the extreme gradient boosting classifier, and the dummy classifier, among others, with a reduction with respect to the multilayer perceptron higher than 10.0, 7.5 and 10.0%, respectively (Table 2).

As the most efficient algorithms, the multilayer perceptron and the decision tree classifier were then used to calculate the generalized weights and thresholds of the physicochemical parameters that discriminated between the *V. dahliae* ID classes, respectively.

The pruned structure of the artificial neural network (multilayer perceptron) contained 12 input nodes (12 physicochemical parameters), three output nodes corresponding to the high, medium, and low *V. dahliae* ID classes, and two hidden layers with 5 and 3 nodes (Figure 5). This was performed using the neuralnet R package (https://CRAN.R-project.org/package=neuralnet, accessed on 2 July 2021) and trained by resilient backpropagation with weight backtracking [32]. The numbers between neurons in Figure 5 represent these weights. The optimal number of hidden layers and nodes was determined by using cross-validation to test the accuracy on the test set.

The second most efficient algorithm (classification and regression trees) was also used to determine the thresholds of the physicochemical parameters that discriminated between the V. dahliae ID classes (Figure 6). This was performed using the XGBoost library for Python [33]. The pruned tree generated by classification and regression trees contained three parameters (clay, lime, and CEC) with four terminal nodes (Figure 6). The percentage of clay was the main factor (i.e., first splitting parameter) that differentiated between the soil samples with a high *V. dahliae* ID class and those in the medium and low ID classes, with a relative percentage of the clay threshold of 31.0%.

The selected neural network and tree were validated with the test set for an independent final check on the accuracy of the model. With the test sets for the neural network, the confusion matrix indicated that few errors were made for the classes ‘medium’ and ‘high’ but some more errors were made for the class ‘low’ (data not shown). This was confirmed by the classification report, which showed a precision of 100.0, 100.0, and 95.0% for the classes high, medium, and low, respectively, with a mean F1 score of 98.0% (Table 3). In the case of the tree classifier, the confusion matrix indicated that few errors were made for the classes ‘low’ and ‘medium’ but some more errors were made for the class ‘high’ (data not shown), as confirmed by the classification report, which showed a precision of 91.0, 94.0, and 50.0% for the classes low, medium, and high, respectively, with a mean F1 score of 90.0% (Table 3).

With the present results, we demonstrated the potential capabilities of machine learning techniques for studying the relationship between the ID of *V. dahliae* in the olive-growing areas of the Iberian Peninsula and the soil physicochemical properties, considering previous cropping history groups as factors. This technology has the potential to be applied for the prediction of epidemics of plant diseases [34]. Between them, the multilayer perceptron network is a supervised learning algorithm which is particularly useful for studying such complex events, as it can capture non-linear relationships of data without explicitly knowing the underlying processes [35]. This algorithm has been satisfactorily used not only to model the pre-planting risk of *Stagonospora nodorum* blotch in winter wheat [35], but also to assess the relationship between Verticillium wilt disease severity and growth indexes in some cotton varieties [36]. In a related case, Zahedi et al. (2016) [36] included one input layer with nine nodes (nine growth indexes), one hidden layer with eight nodes, and one output layer with five foliar disease indexes for modelling. Under our conditions, it was necessary to include 12 input nodes (12 physicochemical parameters), three output nodes corresponding to high, medium, and low *V. dahliae* ID classes, and two hidden layers with five and three nodes, which is not easily understood. Indeed, the interpretation of the weights from this type of machine learning algorithms is difficult; in addition, there is the inability to easily calculate standardized coefficients for each independent variable [37].

The decision tree classifier algorithm had satisfactory results (F1 score higher than 84.0%) in determining the thresholds of the *V. dahliae* colonization, WUE, soil inoculum, or stress parameters that discriminated between the VWO severity classes [14,15]. Additionally, in line with this study, the other machine learning algorithms, such as logistic regression, have also been applied for the prediction of the effects of physicochemical parameters on corn stalk rot [13].

In this study, the decision tree classifier reached a lower F1 score with the test set (90.0%) than the multilayer perceptron network (98.0%) but offered results that were much more easily understood. According to the results from this study, the clay parameter (percentage of clay; %) could be a good indicator for the *V. dahliae* ID level as it was the first splitting parameter that differentiated between the ID classes with a relative percentage of the clay threshold of 31.0%. Curiously, López-Escudero et al. (2010) [4] reported a significantly higher disease incidence in olive orchards established in Vertisol soils or soils with a clay content greater than 30% [38] than that registered in plantations established in soils with a lower clay content (Alfisol).

At the second level, the soils below this clay threshold and with lime < 11.24 meq 100g^−1^ were scarcely *V. dahliae*-infested (low level), while the soil with a higher lime threshold had a medium ID level. In general, lime values lower than 11.24 meq 100g^−1^ were mostly registered in the fields where <2 hosts of *V. dahliae* were cropped in the previous 5 years (barely favourable treatment); so, this parameter could be considered a good indicator as a lower number of hosts were cropped in the field. In this sense, most of the Portuguese fields had lime values lower than 11.24 meq 100g^−1^, which could be considered a good indicator in this area. In addition, the soils with clay above 31.0% and CEC < 40.98 meq 100g^−1^ exhibited a medium ID level, these being the soils separated from the highly infested soils (high ID level) when the CEC was above 40.98 meq 100g^−1^. The soils with these characteristics were exclusively registered in Andalusia (the Cordoba, Seville, and Jaen provinces, Spain) when most of the crops established for five years in the fields were host to *V. dahliae* (very favourable treatment), which rules out CEC as a good indicator under these circumstances.

## 4. Conclusions

The soil infestation of *V. dahliae* in the Guadalquivir Valley, or in other areas where olive groves could be a profitable crop alternative (Castilla La Mancha, Community of Madrid, Portugal, etc.), certainly has important epidemiological consequences, which are already becoming a limiting factor for the expansion of the crop. In this context, there is a complex history of crops and soil properties interactions associated with the increases in the population of *V. dahliae* in the soil [1,12]; so, it is important to uncover the combinations through which both factors generate such increases. The effect of the physicochemical characteristics of the soil—influenced by differential cropping histories—was then assessed by their effect on the ID of *V. dahliae* in the olive-growing areas of the Iberian Peninsula. Other factors affecting the fungus distribution (e.g., irrigation, soil management, and climate) were not considered in this study, and thus, all the findings shown here must be taken with caution in the choice of an orchard for the establishment of the olive.

The highest detection percentage and ID were found in Spain; these values were much lower in the analyzed Portuguese soils, even though the fields with a very favourable previous cropping history had higher ID than did the fields with a barely favourable history, regardless of the origin. The physical parameters of the soil (lime, OM, and the percentage of clay, sand, and silt) and the chemical parameters (CEC, Ca, Mg, Na, K, P, and pH) were able to detect the degree to which the ID was increased by the previous cropping history. In this context, neural networks predict the ID level in soil better than the other algorithms, but the interpretation of the results obtained by this algorithm was complex. By using a decision tree classifier, although it has a lower precision, the percentage of clay was the best indicator for *V. dahliae* ID regardless of the history of the crops. The other parameters can only be used when <2 host crops of *V. dahliae* were established in the field for five years (lime) or four out the previous five years (CEC).

By providing a better understanding of the relationships between the previous cropping histories, the soil properties, and ID, the results found here could show that the soils in the olive-growing areas of the Iberian Peninsula, as established in a previous related studies [4], have acceptable or unacceptable levels for the choice of an orchard for the establishment of the olive.

## Figures and Tables

**Figure 1 jof-08-00988-f001:**
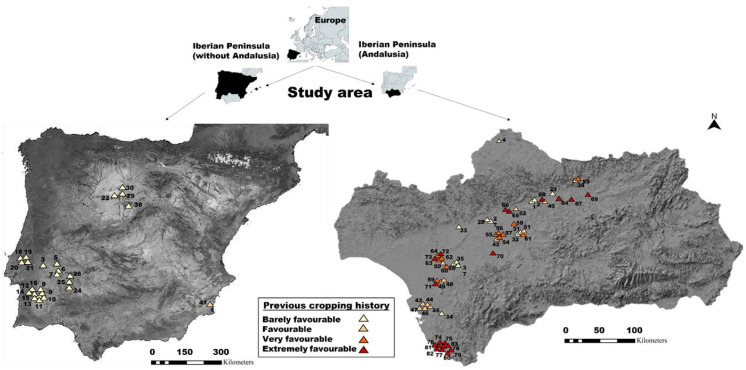
Geographical location of the plots sampled in the Iberian Peninsula. The numbers are the references of the samples (see Appendix A for details). The administrative boundaries were obtained from public spatial data layers of the Environmental Information Network of Andalusia (https://descargasrediam.cica.es/repo/s/RUR?path=%2F) and the Spanish National Geographic Institute (http://www.ign.es). The digital elevation models shown as base maps for the Iberian Peninsula without Andalusia and for Andalusia were downloaded, respectively, from a public database available at (http://www.juntadeandalucia.es/institutodeestadisticaycartografia/prodCartografia/bc/mdt.htm) and (https://www.idee.es:80/csw-inspire-idee/static/api/records/spaignMosaicoLandsat_Historico). Date accessed: 1 July 2021. All the data are compatible with the CC-BY 4.0 license.2.1. Soil sampling and characteristics of surveyed fields.

**Figure 2 jof-08-00988-f002:**
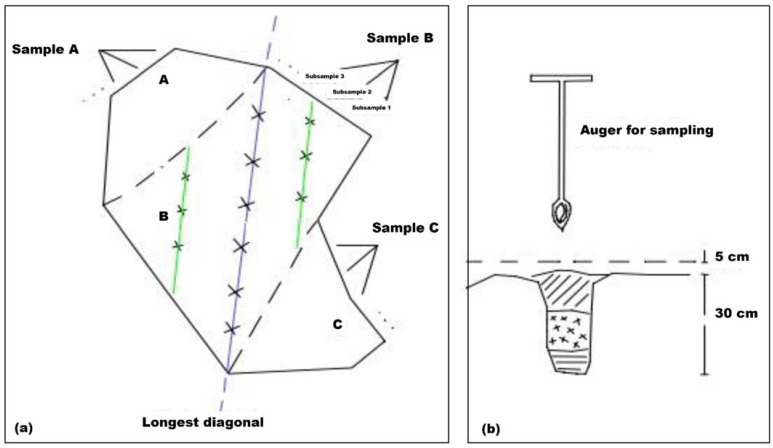
Sampling scheme (**a**) and depth profile (**b**) for the quantification of *Verticillium dahliae* in the soil on plots where olive orchards were to be planted: each sample should be representative of a homogeneous surface for the type of soil, type of cultivation, or type of cultural practices and not greater than 2–2.5 ha. If the homogeneous plot was larger or had different types of soil or crop rotations, it was divided into smaller sampling units. In this way, the initial plot was divided into several subplots (A, B, C, etc.), from which a sample was obtained: sample A, sample B, sample C, etc. Each of the samples was in turn composed of different subsamples (1, 2, 3, etc.) that had to be taken every 10-20 m (green lines) on the longest diagonal (purple line) of each of the subplots.

**Figure 3 jof-08-00988-f003:**
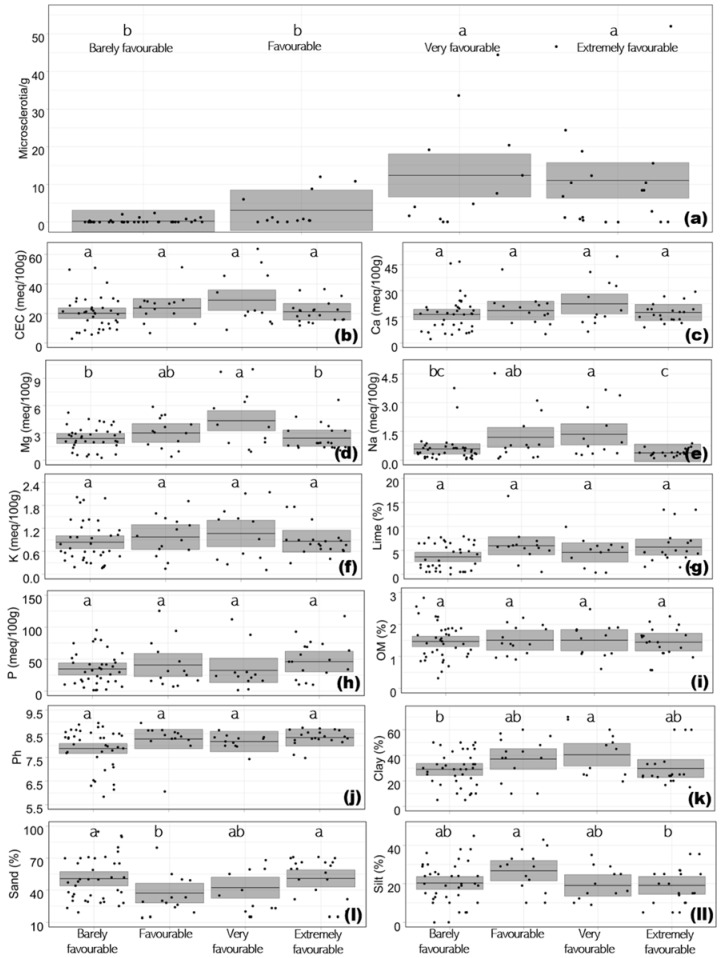
Raw values of inoculum density of *Verticillium dahliae* in soil (**a**) and cation exchange capacity (CEC; (**b**)), Ca exchange (Ca; (**c**)), Mg exchange (Mg; (**d**)), Na exchange (Na; (**e**)), K exchange (K; (**f**)), assimilable P (P; (**g**)), active limestone (lime; (**h**)), organic matter (OM; (**i**)), pH (**j**), and percentage of clay (**k**), sand (**l**), and silt (**ll**) raw values from 84 samples collected in the Iberian Peninsula and the mean from the linear mixed model with 95% confidence intervals per previous cropping history group (VWO which was barely favourable with <2 host crops to *V. dahliae* in the previous five years; VWO which was favourable with 3 host crops of the fungus; VWO which was very favourable and indicated that only one of the previous five years of the plot was free of *V. dahliae* hosts; and VWO which was extremely favourable and indicated that during all the five previous years species susceptible to *V. dahliae* were cropped in the plot) and grouping letters (Treatments with a common lowercase letter do not differ significantly according to Tuckey test at *P* = 0.05).

**Figure 4 jof-08-00988-f004:**
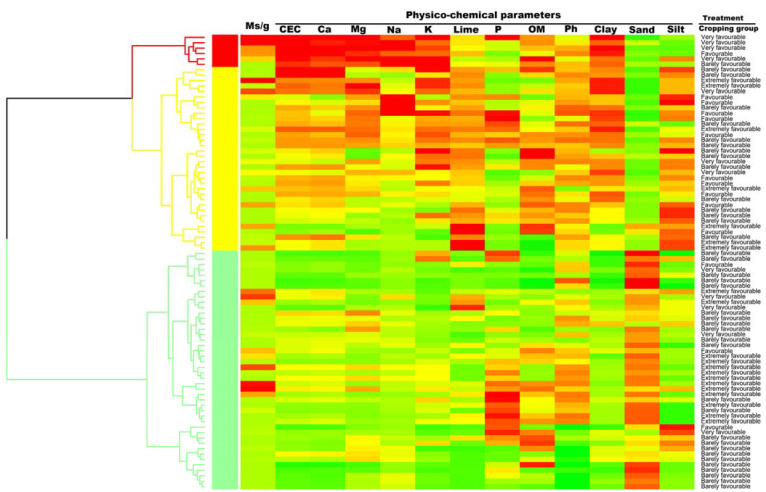
Dendrogram derived from cluster analysis of 4 experimental treatment combinations in a study where the effect of previous cropping history was assessed on densities of *Verticillium dahliae* in olive-growing areas of the Iberian Peninsula and heat map representation of the ID and soil physicochemical parameters. The 12 parameters selected for the heat map representation were CEC (meq 100g^−1^), Ca (meq 100g^−1^), Mg (meq 100g^−1^), Na (meq 100g^−1^), K (meq 100g^−1^), P (meq 100g^−1^), lime (%), OM (%), pH, and percentage of clay, sand, and silt. Agglomerative cluster analyses were performed based on the Spearman correlation matrix calculated from values of the different parameters using the Ward method. Clusters of experimental treatment combinations, depicted in different colors, were estimated on the basis of the average silhouette width according to the Mantel statistic. In the heat map, in each column, cells represent the value of each parameter for each experimental treatment combination of four previous cropping history groups (VWO with a barely favourable previous cropping history, with <2 host crops of *V. dahliae* in the previous five years; VWO with a favourable previous cropping history, with 3 host crops of the fungus; VWO with a very favourable previous cropping history which indicated that in only one of the previous five years the plot was free of *V. dahliae* hosts; and VWO with an extremely favourable previous cropping history which indicated that during all the five previous years the species susceptible to *V. dahliae* were cropped in the plot) from 84 samples.

**Figure 5 jof-08-00988-f005:**
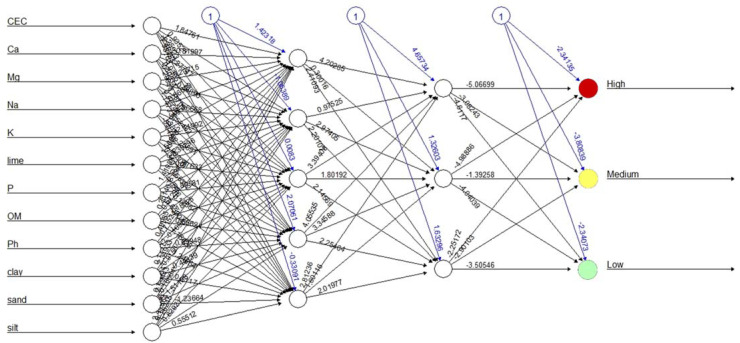
Pruned structure of the multilayer perceptron artificial neural network for predicting *Verticillium dahliae* inoculum density (ID) classes (low, medium, and high) in a study where the effect of cropping systems was assessed on the ID of *V. dahliae* in olive-growing areas of the Iberian Peninsula based on CEC (meq 100g^−1^), Ca (meq 100g^−1^), Mg (meq 100g^−1^), Na (meq 100g^−1^), K (meq 100g^−1^), P (meq 100g^−1^), lime (%), OM (%), pH, and percentage of clay, sand, and silt parameters. The neural net was trained by resilient backpropagation with weight backtracking [32] reaching two hidden layers of 5 × 3 neurons as optimal. The numbers between neurons represents these weights. Data include a training set (75%) from a dataset (84 samples) comprising all experimental combinations of four previous cropping history groups (VWO—barely favourable, favourable, very favourable, and extremely favourable).

**Figure 6 jof-08-00988-f006:**
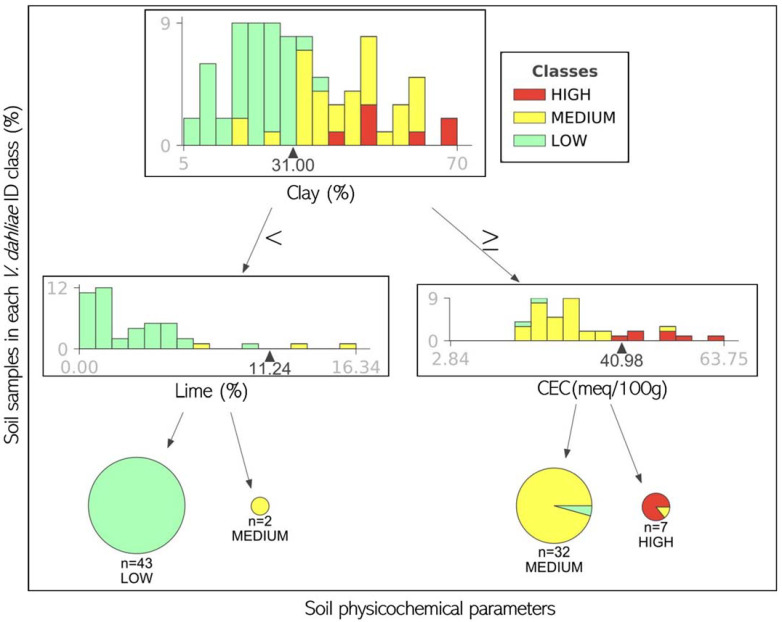
Pruned classification tree for predicting *Verticillium dahliae* inoculum density (ID) classes (low, medium, and high) in olive-growing areas of the Iberian Peninsula based on CEC (meq 100g^−1^), Ca (meq 100g^−1^), Mg (meq 100g^−1^), Na (meq 100g^−1^), K (meq 100g^−1^), P (meq 100g^−1^), lime (%), OM (%), pH, and percentage of clay, sand, and silt parameters. The histogram for each node represents the percentage of soil samples in each *V. dahliae* ID class. For each terminal (decision) node, the most prevalent *V. dahliae* ID class is indicated showing feature distributions as overlapping stacked histograms, one histogram per target class. Leaf size is proportional to the number of samples in that leaf. Data include a training set (75%) from a dataset (84 samples) comprising all experimental combinations of four previous cropping history groups (VWO—barely favourable, favourable, very favourable, and extremely favourable).

**Table 1 jof-08-00988-t001:** Plant species cropped in sampled fields in the five years before soil analyses.

Name	Family	Species	Host Crops to *V. dahliae* ^a^
Artichoke	Asteraceae	*Cynara scolymus*	+
Barley	Poaceae	*Hordeum vulgare*	−
Bean	Fabaceae	*Vicia faba*	−
Broccoli	Brassicaceae	*Brassica oleracea var. italica*	−
Carrot	Apiaceae	*Daucus carota*	+
Chickpea	Fabaceae	*Cicer arietinum*	−
Corn	Poaceae	*Zea mays*	−
Cotton	Malvaceae	*Gossypium hirsutum*	+
Garlic	Amaryllidaceae	*Allium sativum*	−
Grapevine	Vitaceae	*Vitis*	−
Leek	Amaryllidaceae	*Allium ampeloprasum var. porrum*	−
Lettuce	Asteraceae	*Lactuca sativa*	+
Lucerne	Fabaceae	*Medicago sativa*	+
Melon	cucurbitáceas	*Cucumis melo*	+
Oak	fagáceas	*Quercus ilex*	−
Oats	Poaceae	*Avena sativa*	−
Olive	Oleaceae	*Olea europaea*	+
Onion	Amaryllidaceae	*Allium cepa*	−
Peach	Rosaceae	*Prunus persica*	+
Potato	Solanaceae	*Solanum tuberosum*	+
Pumpkin	cucurbitáceas	*Cucurbita pepo*	+
Raygrass	Poaceae	*Lolium hybridum Hausskn*	−
Rice	Poaceae	*Oryza sativa*	−
Sugar beet	Amaranthaceae	*Beta vulgaris*	+
Sunflower	Asteraceae	*Helianthus annuus*	+
Sweet potato	Convolvulaceae	*Ipomoea batatas*	+
Tomato	Solanaceae	*Solanum lycopersicum*	+
Triticale	Poaceae	*Triticum aestivum + Secale cereale*	−
Wheat	Poaceae	*Triticum aestivum*	−

^a^ The species were noted as hosts of *Verticillium dahliae* according to information reviewed by Pegg and Brady (2002) [1].

**Table 2 jof-08-00988-t002:** Statistical parameters obtained from different linear and nonlinear machine learning algorithms in classifying ‘low’, ‘medium’, and ‘high’ *Verticillium dahliae* ID classes with several physicochemical parameters in a study where the effect of previous cropping history was assessed on densities of *V. dahliae* in olive-growing areas of the Iberian Peninsula.

Model	Accuracy	Balanced Accuracy	F1 Score	Time Taken
Multilayer perceptron	0.952	0.889	0.949	0.012
Decision tree classifier	0.905	0.856	0.901	0.022
Ridge classifier CV	0.905	0.856	0.901	0.015
Support vector machines	0.905	0.856	0.901	0.011
Extra trees classifier	0.905	0.856	0.901	0.159
Nearest centroid	0.905	0.856	0.901	0.008
Label propagation	0.905	0.856	0.901	0.021
Label spreading	0.905	0.856	0.901	0.012
Ridge classifier	0.857	0.822	0.854	0.020
Linear discriminant analysis	0.857	0.822	0.854	0.016
Stochastic gradient descent classifier	0.857	0.822	0.854	0.015
K-neighbors classifier	0.857	0.822	0.854	0.022
Gaussian naive Bayes	0.857	0.744	0.838	0.019
XGBClassifier	0.810	0.711	0.791	0.085
Bernoulli naive Bayes	0.810	0.711	0.797	0.013
Linear support vector classifier	0.810	0.711	0.791	0.018
Logistic regression	0.762	0.678	0.743	0.018
Light gradient boosting machine classifier	0.810	0.633	0.748	0.073
Passive-aggressive classifier	0.810	0.633	0.748	0.017
Calibrated classifier CV	0.810	0.633	0.748	0.036
Random forest classifier	0.762	0.600	0.701	0.214
Bagging classifier	0.762	0.600	0.701	0.034
Adaptive boosting classifier	0.714	0.558	0.660	0.083
Quadratic discriminant analysis	0.667	0.533	0.606	0.017
Extreme gradient boosting classifier	0.619	0.500	0.569	0.025
Dummy classifier	0.286	0.225	0.273	0.000

**Table 3 jof-08-00988-t003:** Classification report of a multilayer perceptron artificial neural network and a decision tree classifier directly run on the test set to estimate the relationship between the classes of *Verticillium dahliae* inoculum density in olive-growing areas of the Iberian Peninsula and several physicochemical parameters.

	Artificial Neural Network				Decision Tree			
	Precision	Recall	F1 Score	Support	Precision	Recall	F1 Score	Support
low ^1^	0.95	1.00	0.94	21	0.91	1.00	0.95	21
medium ^1^	1.00	0.90	0.95	19	0.94	0.84	0.89	19
high ^1^	1.00	0.67	1.00	2	0.50	0.50	0.50	2

accuracy	-	-	0.98	42	-	-	0.90	42
macro avg	0.98	0.98	0.98	42	0.78	0.78	0.78	42
weighted avg	0.98	0.98	0.98	42	0.91	0.90	0.90	42

^1^ ID level of *V. dahliae* in soil.

## Data Availability

All data analyzed in this study are included in this article and its Appendix A.

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
