# Peer review of "Effect of Previous Crops and Soil Physicochemical Properties on the Population of Verticillium dahliae in the Iberian Peninsula"

_jof, 2022, doi:10.3390/jof8100988_

Round 1

Reviewer 1 Report

The anuscripst you presented is really well written and shows a good job done to go further on the study of the V. dahliae in such an important crop as olive trees.

This is the first study that relates soil physicochemical properties-cropping systems relationship in relation to the ID of V. dahliae in soil. This opens an important tool to decide to stablish new plantations or not.

There is small error in page 1, line 35, a missing dot after etc, and in page 2, line 58, where I think that the name of the uthor needs to be written to have an understandable read.

Congratulations.

Reviewer 2 Report

This manuscript written by Antonio et al. analysed the effect of crop history and edaphic factors on Verticillium dahliae population in the soil of the Iberian Peninsula.

It is very interesting study to have a wide range of soil collections and analysis to predict the V. dahliae inoculum density in the soil. It provides very useful tools to select fields to establish olive plants.

Line 20: rephrased this sentence.

Line 58: by [10] seems a bit strange, better to put the author’s name.

Line 58-60: rephrase this sentence, using normal English language to describe the treatment instead of some symbols, e.g. “+”. What is “high N”, “high Na” and “low K”? 

“In olive”? Do you mean “in the case of olive plants/trees”?

“Inoculation of plants”?

After reading paragraph starting in line 62, would suggest to move “soil texture” in line 49 after “pH” and “cation exchange capacity”.

Line 70: “effect of irrigation on the disease”? on disease occurrence or disease suppression? Be a bit more specific and precise in the description.

Line 81: “relationship mentioned above”? which relationship among/between which parameters?

Line 91-92: “former analyses”? Which analyses? For determining ID of V. dahliae? “The second ones”? it’s better to write what are those second ones.

Figure 2 is a very informative figure to show the soil sampling process, but it can be more organized, for instance to separate left and right into two panels, 

Figure 3 can be improved in the presentation. The legend of y-axis of each panel can be more precise with unit or a short description. The lowercase letters which were used to indicate statistics can be moved on the top of the boxplots, and usually “a” indicates the highest values. The colors of points and boxplots are not very visible, especially for “Barely favourable”. Actually, it’s not very necessary to use different colors to indicate different groups, just easily put the names of groups on the x-axis.

Figure 4 need to be improved. The text under treatment/cropping group on the right of the figure are invisible, they are too small and too “colorful”, please revise it to improve the visibility. Or use an additional table to summarize the right side of the figure?

Figure 5 needs to be improved. Too many texts (numbers) in the figures, probably remove most of the unimportant numbers and highlight the important numbers.

Figure 6 can be improved with separating the panels, and add legends of y-axis and x-axis.

I am not very familiar with the article structure requirement of Journal of Fungi, it’s probably better to separate result section and discussion section.

I was wondering whether there are any geographic patterns of V. dahliae inoculum density in the soil? Are there any (micro)climate differences, e.g. temperature, precipitation, among different sampling sites?

For future study, it will be very interesting to investigate the effect of soil microbiome changes on V. dahliae.

This is in general well written manuscript, but there are some sentences seems a bit confusing. I highlighted them in yellow background in the pdf file.
